# Characterization of Evolution Stages, Groundwater and Soil Features of the Mud Forest Landscape at Qian-an (China)

**XiangJian Rui, Lei Nie, Yan Xu \*, Chao Du, FanSheng Kong, Tao Zhang, YuanYuan He and YuZheng Wang**

Construction Engineering College, Jilin University, Changchun 130026, China; ruixj17@mails.jlu.edu.cn (X.R.); nielei@jlu.edu.cn (L.N.); duchao18@mails.jlu.edu.cn (C.D.); kongfs19@mails.jlu.edu.cn (F.K.); zhangtao18@mails.jlu.edu.cn (T.Z.); hyy20@mails.jlu.edu.cn (Y.H.); yuzhengw18@mails.jlu.edu.cn (Y.W.)
\* Correspondence: xuyan8102@jlu.edu.cn

**Abstract:** The research on geological landscape has received more and more attention worldwide. The National Geological Park of Qian-an mud forest, located in Qian-an Country, Songyuan City (Jilin Province, China) is a rare natural geological landscape formed by erosion. Mud forest landscape has undergone long-term geological processes, and it is still in continuous evolution due to subsurface erosion. In the process of the mud forest landscape formation and evolution, distinct stages have been recognized. The subsurface erosion factors of the mud forest area were identified by groundwater and soil samples characterization, and the mechanism of the formation of the mud forest is studied. Results show that the occurrence of subsurface erosion is controlled by four factors: (1) The head difference of terrace increases due to geological structure, (2) The dry and cold paleoclimate increases the accumulation of soluble salts. Concentrated precipitation in the short term also promotes subsurface erosion. (3) The high content of sodium ions in groundwater promotes the dispersion of soil, and (4) Loess-like soil is characterized by high porosity, low plasticity, and dispersibility.

**Keywords:** mud forest; erosion; subsurface erosion; landscape formation; landscape evolution; loess-like soil

## 1. Introduction

Soil erosion is a global problem and poses major issues in many countries [1,2]. Soil erosion can increase run-off by changing soil properties and particularly by destroying topsoil structure and reducing water holding capacity [3,4]. Soil erosion is not only a land degradation process, but also a geomorphological appearance [5]. When the rate of soil erosion exceeds the rate of soil formation, the process of land degradation helps shaping the natural geological landscape [6]. In other words, while soil erosion causes land degradation, it also forms some typical landscapes (Table 1). Kuhn, et al. [7] studied the effect of rainfall on the Zin Valley badlands in Israel. "Calanchi" and "biancane" are two typical badland landscapes in Italy. Ciccacci, et al. [8] analyzed the evolution of "biancane" landforms, whereas Pulice, et al. [9] investigated how topography and rainfall may affect runoff and slope processes in "calanchi" landforms. Most of the soil erosion landscapes studied above are gully shaped or round enveloped landscapes formed by surface erosion. The research object of this paper is a columnar geological landscape formed by both surface and subsurface erosion. As seen in Table 1, the studied landscapes are mainly composed of earth pillars, arranged together looking like a forest, hence it was named "mud forest" [10,11].

**Table 1.** Several typical geological landscapes formed by soil erosion. [7–9,12,13].

| Name | State | Picture |
|------|-------|---------|
| Zin Valley badlands | Israel |  |
| Badland landscape | Spain |  |
| "Calanchi" badland landform | Italy |  |
| "Biancane" badland landform | Italy |  |
| Yuanmou Dry-hot Valley | China |  |
| Mud forest landscape | China |  |

The process of erosion is caused by both surface and subsurface processes. Over the last decades, most studies on soil erosion have focused on surface processes, such as sheet, rill and gully erosion [5,14]. Although subsurface erosion has been reported to be a significant and widespread process, the disproportion in the number of studies on surface erosion compared to those on subsurface erosion is striking [5,15]. Subsurface erosion generally refers to various forms of erosion caused by groundwater below the surface [16]. This process and its effects have been described under a variety of names: subterranean erosion [17], subcutaneous erosion [18], sinking of the ground [19–21], sink-hole erosion [22,23], tunnel erosion [24], pothole gullying [25], tunnel-gully erosion [26,27], tunneling erosion [28], piping [29], soil piping [30], pothole erosion [31], and suffusion [32–34]. While degradation occurs in all kinds of landscapes over the world, the drivers of degradation vary from region to region [35]. Due to the complexity of subsurface erosion, no single factor can be held responsible for subsurface erosion development [5]. Vannoppen, et al. [36] pointed out that the

occurrence of subsurface erosion is controlled by soil and climate features [37–41]. The occurrence of subsurface erosion is also closely related to topography [42–45].

In 2009, the Qian-an mud forest landscape was officially approved as a national geological park in China. Due to its unique landform caused by erosion, this landscape has become the only protected "mud forest" site in China with these geological features, giving the area a high aesthetic and scientific investigation value [46]. However, mud forest is a very fragile geological landscape and suffers from serious soil erosion. In order to maintain mud forest stable for a long time, it is necessary to understand the causes of such landscape degradation. Previous researches on mud forest landscape have been carried out, including the formation mechanism, development and utilization, slope failure. Zhu and Liang [47] preliminarily analyzed the formation mechanism of the mud forest landscape. Zhou, Wang, Ren, Cai and Zhang [10] believed that the formation of the mud forest landscape was related to landform and wind force. On the basis of previous studies, Chi, Wang and Yang [11] proposed that the formation and development of mud forests were not only related to external factors, but also related to soil composition. Zhang et al. [48] concluded that the frost weathering is an important factor in the damage in the region. However, the formation mechanism of the mud forest has not been clearly analyzed since the influence of interflow was ignored in previous studies. In order for this landscape to remain stable for a long time, it is necessary to understand the causes of landscape evolution.

Therefore, this study focused on the role of subsurface erosion in the formation and evolution of the mud forest landscape, by using different evolution stages to describe such landscape development more intuitively. At the same time, the factors affecting the subsurface erosion in mud forest area were analyzed in details in order to provide theoretical basis for landscape protection.

## 2. Materials and Methods

### 2.1. Study Area

The National Geological Park of Qian-an mud forest is located in the southwest of Qian-an Country, Songyuan City, Jilin Province (Figure 1). It is 40 km away from Qian-an Country, across Suozi Town and Dabusu Town. The coordinates are longitude 123°36′–123°42′ and latitude 44°45′–44°50′, with a total area of 110 km². Mud forest scenic spot is located in the Dabusu nature reserve, and the area of the mud forest landscape is 7.5 km².

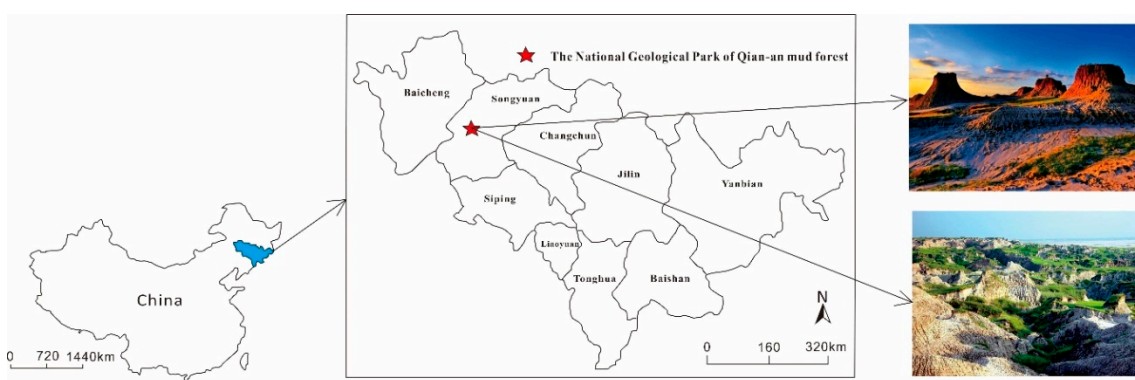

**Figure 1.** Location of the National Geological Park of Qian-an mud forest and mud forest landscape.

The study area has a continental arid and semi-arid monsoon climate in the north temperate zone. This climate zone presents the distinctive climate characteristics of dry and windy springs, warm and rainy summers, cool and short autumns, and long and cold winters. The perennial average temperature in the area is 4.7 °C. January records the lowest low temperature at −14.8 °C, while in July the temperature is 24.9 °C. Average annual precipitation is about 400–500 mm and two-thirds of this amount occurs in June, July, and August. The average sunshine duration reaches 2900 h within a year. The frost-free days are 145 per year. The vegetation in the study area is characterized by

meadow grassland. In the park there are 274 species of plants in 51 families, including huangying, sedge, hornwort, and *achnatherum splendens*.

The study area is distributed in the western alluvial plain of the Songnen Plain. The terrain is sloped from east to west and from south to north due to neotectonic movement. The elevation is 150–170 m in the east and 140–155 m in the west. This results in no transit rivers in the Qian-an area. However, a series of lake depressions closely related to the ancient channel are left, forming a relatively independent closed-stream area. In the west of the Songnen Plain, there are many lakes and marshes. Preliminary statistics shows that nearly 700 lakes exist within an area of over 6 km$^2$. Dabusu Lake is a large inland lake among these lakes [10]. Because of the hot weather in summer, the evaporation is much larger than precipitation, which causes a large accumulation of saline and alkaline substances near Dabusu Lake. Dabusu Lake has a low altitude and it is the center of groundwater gathering. Dabusu Lake receives the joint replenishment of atmospheric precipitation, surface runoff and subsurface current. The catchment area is nearly 230 km$^2$, and the lake basin area is about 81 km$^2$. As shown in Figure 2, there are 10 gullies in the west and 4 gullies in the east of Dabusu Lake.

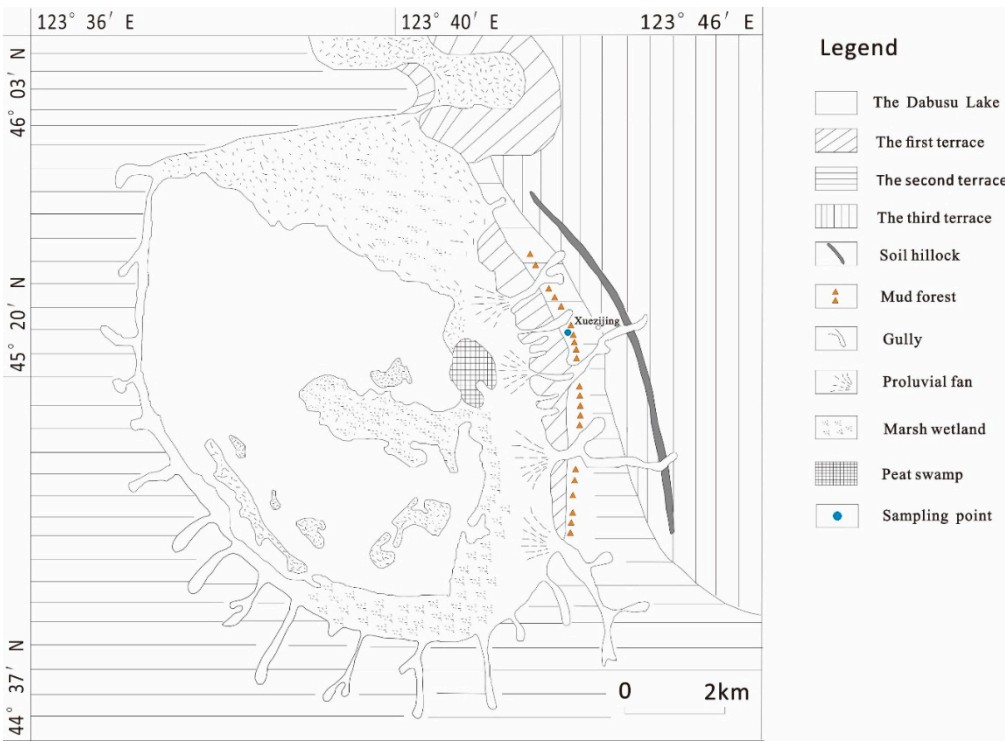

**Figure 2.** Dabusu Lake area plan.

The Songnen Plain, where the Dabusu Lake is located, is a basin type plain that has been continuously settled since Mesozoic and Cenozoic era and accumulated huge thickness of Mesozoic sediments. The basement of Songnen Plain is the metamorphic rock series of the pre-Jurassic period, and the caprock is the meso-Cenozoic deposit with a thickness of 4000–5000 m. According to relevant data, the stratigraphic lithology of Qian-an area is shown in Table 2 [10]. Dabusu mud forest geomorphologic landscape was produced in Guxiangtun formation (upper Pleistocene), consisting in silt, silty clay and clay. The mud forest area is located in the secondary terrace on the east side of Dabusu Lake (Figure 2). This area is located in the middle of the central sag of the secondary structure of the Songliao giant subsidence zone. Soils in the study area are characterized by poor profile differentiation and the area is classified as Cambisols according to the World Reference Base for Soil Resources (WRB) [49].

**Table 2.** Stratigraphic lithology of Qian-an area [10].

| Geologic Symbols | Chronolithologic Unit | Lithostratigraphic Unit | Burial Depth (m) | Formation Thickness (m) | Lithology |
|---|---|---|---|---|---|
| Q4 | Quaternary Holocene | - | 0–1 | 0–1 | Cultivated soil |
| Q3g | Quaternary upper Pleistocene system | Guxiangtun formation | 1–18 | 15–34 | Loess-like soil |
| | | | 18–34 | | Fine sand |

The mud landscape geomorphologic group is mainly distributed in the leading edge of terraces, and the earth pillar height is 10–15 m. Figure 3 shows the geological evolution diagram of the mud forest landscape area. On the basis of the formation process of Mesozoic, early Pleistocene and middle Pleistocene, the area experienced the late Pleistocene lake paleogeographic depositional environment. The formation in this area is characterized by loess-like soil in the upper part and fine sand in the lower part. After the early late Pleistocene deposition, the crust in this area underwent the process of uplift in the east and depression in the west and inclination in the northwest in Neoid period. In the middle of late Pleistocene, due to the uplift of the crust in the Neoid period, erosion and subsurface erosion are intensified, thus forming the characteristic geological and geomorphic landscape of the mud forest. As subsurface erosion continues, the landscape of the mud forest is still changing.

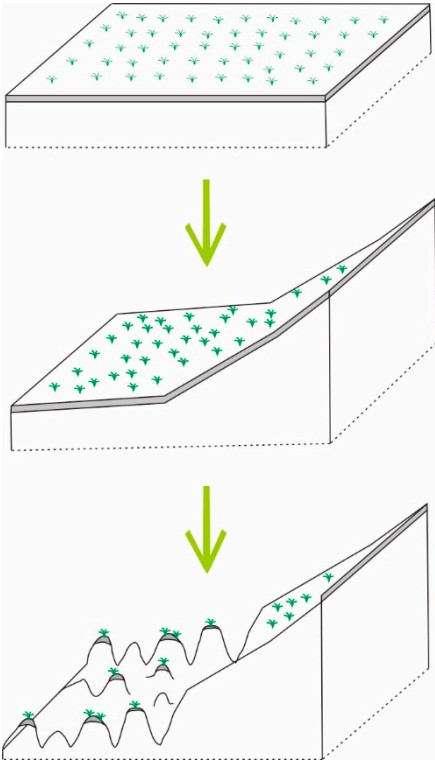

**Figure 3.** A schematic geological evolution diagram of the mud forest landscape area.

*2.2. Field Survey and Laboratory Test*

A detailed field survey was conducted in the study area in order to assess different evolution stages of the mud forest landscapes. In order to study the influence of soil on the subsurface erosion in mud forest area, three soil samples were collected between soil pillars at a depth of 10 cm, namely C1, C2, and C3. Three groundwater samples were also collected for experimental study, namely N1, N2, and N3 (Figure 2). The main ions content of groundwater and soluble salt in soil samples were measured. The test methods are shown in Table 3. The groundwater chemistry was analyzed

by piper diagrams and categorized by the Gibbs' model. In the piper diagrams in Figure 4a: (i) In region 1, alkali earth metal ions are greater than alkali metal ions. (ii) In region 2, alkali metal ions are greater than alkali earth metal ions. (iii) In region 3, weak acid root ions are greater than strong acid root ions. (iv) In region 4, strong acid root ions are greater than weak acid root ions. (v) In region 5, the carbonate hardness is greater than 50%. (vi) In region 6, the noncarbonate hardness is greater than 50%. (vii) In region 7, noncarbonate bases are greater than 50%. (viii) In region 8, carbonate base is greater than 50%. (ix) In region 9, no pairs of cation-anion were greater than 50%. Based on the analysis of water in the world's rivers, lakes and major oceans, Gibbs [50] believes that the controlling factors of ion origin can be divided into rock weathering type, atmospheric precipitation control type and evaporation-concentration type. Figure 4b shows the distribution principle of the Gibbs model.

**Table 3.** Test method for chemical composition of groundwater and soluble salt in soil samples.

| Test Item | Method |
| --- | --- |
| $HCO^-_3$ | Double indicator neutralization |
| $Cl^-$ | Silver nitrate titration |
| $SO_4^{2-}$ | Mass method |
| $Ca^{2+}$, $Mg^{2+}$ | EDTA coordination titration |
| $Na^+$, $K^+$ | Flame photometry |
| pH | pHS-3C tester |

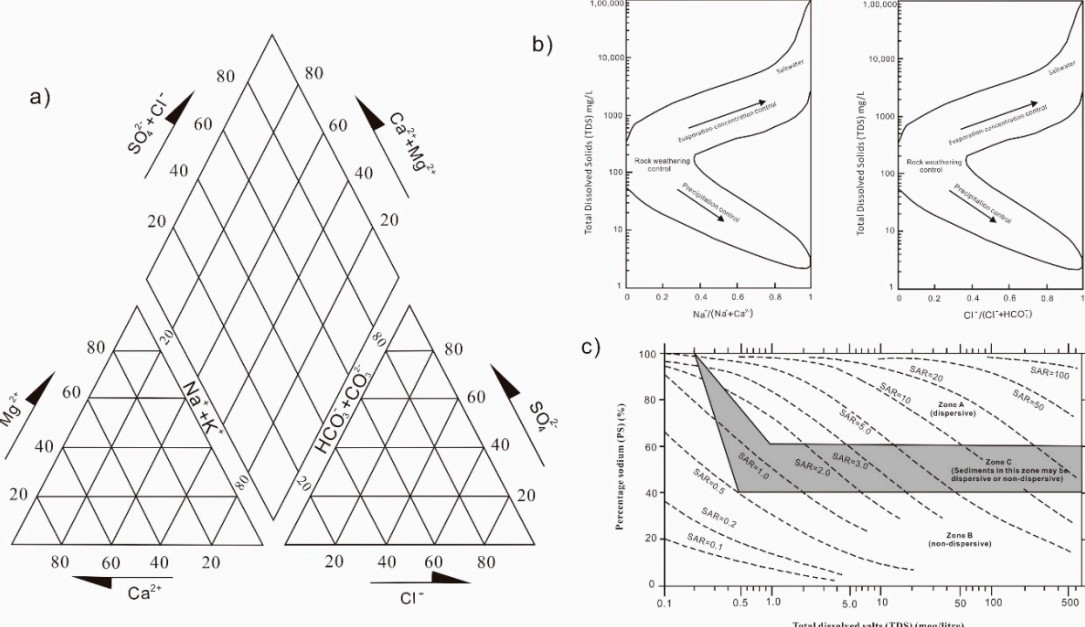

**Figure 4.** Methods used in the experimental analysis: (**a**) Piper diagrams. (**b**) Gibbs model. (**c**) Sherard diagram.

Grain size distribution was measured by a combination of sieving and hydrostatic sedimentation methods. The natural density was measured by cutting ring method. Natural moisture content was measured by means of drying. Liquid limit and plastic limit were determined by the combined determination of liquid limit and plastic limit. Porosity was measured by mercury injection. Major chemical elements were determined via X-ray fluorescence spectrometry following the analytical procedure of Franzini et al. [51] and Leoni and Saitta [52]. This method uses powder pellets and is based on the full matrix correction method. Total volatile components ($H_2O$ and $CO_2$) were determined as loss on ignition (LOI) at 950 °C on powders dried at 105 °C [53–55]. Considering the high ion content of the soil sample, the double hydrometer method was not suitable, so the dispersibility of the soil sample was determined by Sherard diagram [56]. Several parameters are necessary to determine whether the soil has

dispersibility [57,58], among them: the "sodium adsorption ratio", SAR = $Na^+/[(Ca^{2+} + Mg^{2+})/2]1/2$; the "percentage sodium", PS = $[Na^+/(Na^+ + K^+ + Ca^{2+} + Mg^{2+})] \times 100$; the "total dissolved salts", TDS = $Na^+ + K^+ + Ca^{2+} + Mg^{2+}$. Sherard, Dunnigan and Decker [56] correlated TDS, SAR, PS and clay dispersibility (Figure 4c), according to these authors, the clays of zone A have a high tendency for spontaneous dispersion, the sediments of zone C may be dispersive or nondispersive and the materials of zone B are ordinary erosion-resistant clays [55].

## 3. Results and Discussion

### 3.1. Mud Forest Landscape Fetaures and Evolution Stage

According to the field investigation, it was found that the mud forest landscape in the study area is characterized by distinct stages, named infant stage, juvenile stage, youth stage and old stage. These four stages reflect to the germination, formation, development and extinction stages of the mud forest landscape (Figure 5).

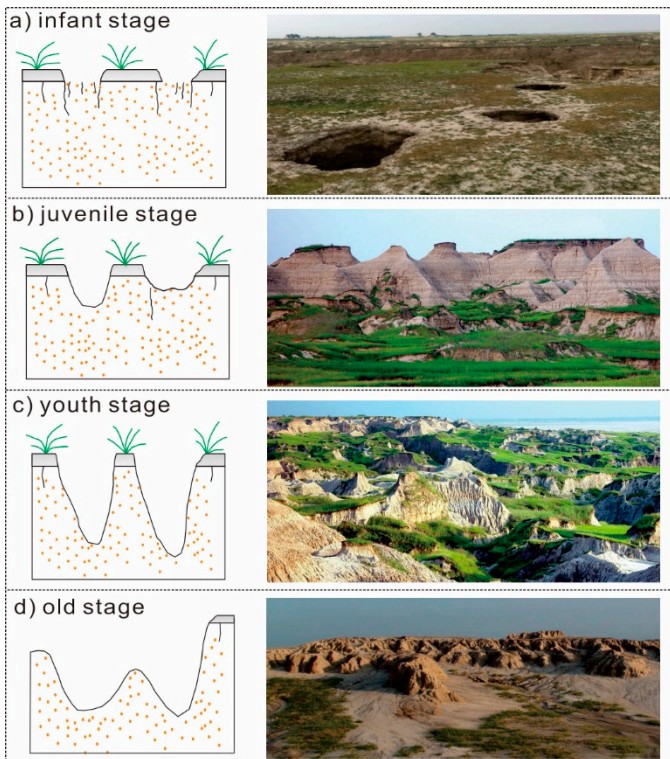

**Figure 5.** Geomorphologic evolution and different periods of the mud forest landscapes: (**a**) the infant stage; (**b**) the juvenile stage; (**c**) the youth stage; and (**d**) the old stage.

In the infant stage, because of surface erosion, small patches of cultivated soil are destroyed and the vegetation cannot grow. This exposed the loess-like soil directly (Figure 6a). First, the lack of cultivated soil layer makes terrain concave. Precipitation tends to accumulate there. Second, due to the lack of vegetation protection, the exposed loess-like soil is more conducive to rainfall infiltration. Precipitation penetrates along pores and vertical joints (Figure 5a). The eluviation makes the joints and fissures of loess-like soil enlarge. During long years of erosion, small cracks expand to form caves. Due to the special physical and chemical properties of loess-like soil, it is easy to disintegrate and dissolve when meeting water. This causes the soil layer to be vulnerable to subsurface erosion, thus expanding the cave range. Under the action of gravity, the top layer of soil collapses, creating holes (Figure 6b). With the continuation of the subsurface erosion, the sinkholes and the horizontal suffusion caves are developed. Some vertical sinkholes have also been formed (Figure 6c).

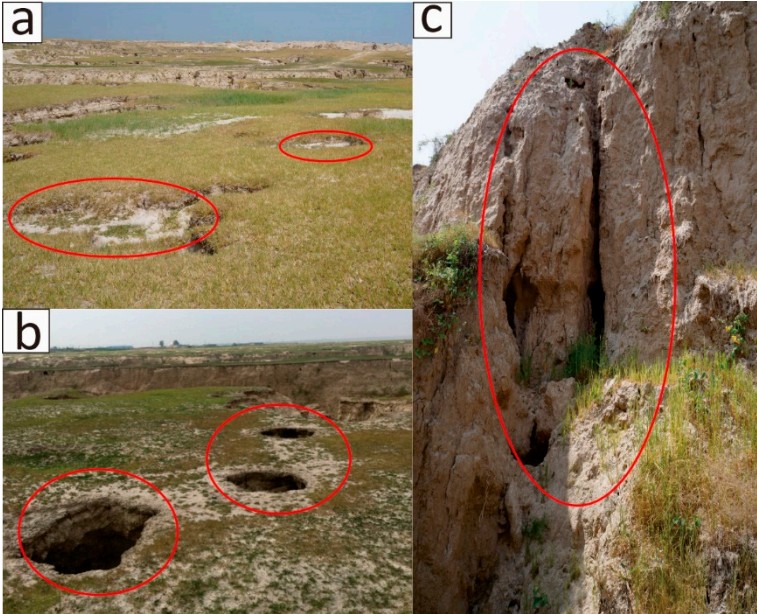

**Figure 6.** Geomorphologic features of the infant stage of mud forest evolution. (**a**) The lack of vegetation on a small area of the earth's surface leaves the loess-like soil exposed; (**b**) Many holes are shaped by subsurface erosion; (**c**) Vertical sinkhole are formed by subsurface erosion.

In the next juvenile stage, a large number of holes were formed. With the passage of time and the continuous occurrence of erosion, these holes expanded leading to a destruction of terraces integrity. Gradually, gullies of different sizes and shapes formed, showing accordingly the form of "island" (Figure 5b).

With further development of dissolution and subsurface erosion, the ornamental value of the mud forest reaches its peak in the youth stage of landscape development. The solitary peaks are irregular and varied in shape. Some look like conical and humped forms (Figure 5c).

After reaching the peak of mud forest landscape development, subsurface erosion continues to occur. However, continued subsurface erosion will begin to destroy the mud forest landscape. The mud forest landscape gradually loses its charm and looks like a series of small raised planes from a distance (Figure 5d). Although its form is unique, its ornamental value is far less than the youth stage of the mud forest landscape. This is the old stage of the mud forest landscape, it is also a stage of landscape degradation. At present, most of the mud forest landscape is in this stage.

*3.2. Groundwater Analyses*

The milligram equivalent percentage of each ion was calculated according to the content of each component of the groundwater (Table 4). It is easy to see that sodium ions account for the largest milligram equivalent percentage of cations, at 55.31%. Calcium ions were the second, accounting for 35.91%. Among the anions, chlorine ions accounted for 55.22%, which was the largest proportion of anions. The bicarbonate ions accounted for 37.92%. Water sample was analyzed by Piper diagrams [57,58]. The sample plots fall under zone 2, 4, and 7 (Figure 7). According to the Shukalev classification method, the groundwater chemistry in the study area is of type $HCO_3$ + Cl-Na + Ca. The calculated $Na^+/(Na^+ + Ca^{2+})$ value is greater than 0.6 and $Cl^-/(Cl^- + HCO^-_3)$ value is close to 0.5. The groundwater test data was plotted in the Gibbs model. The water sample plots the slightly upper right side of the model center (Figure 8). This reflects that the water characteristics in this region are controlled by evaporation and crystallization.

**Table 4.** Physical and chemical properties and main ion content of groundwater.

| Test Item | Value (mg/L) | Test Item | Value (meq/L) | Milligram Equivalent Percentage (%) |
|---|---|---|---|---|
| Visible substances | White and turbid | $SO_4^{2-}$ | 2.065 | 6.86 |
| Water temperature | 15 (°C) | $Cl^-$ | 16.617 | 55.22 |
| pH | 7.01 | $HCO^-$ | 11.411 | 37.92 |
| ORP | 127.1 | $Na^+$ | 20.955 | 55.31 |
| Water hardness | 842.8 | $K^+$ | 0.018 | 0.047 |
| Total dissolved solids | 2184 | $Ca^{2+}$ | 13.605 | 35.91 |
| Water alkalinity | 571.1 | $Mg^{2+}$ | 3.307 | 8.73 |

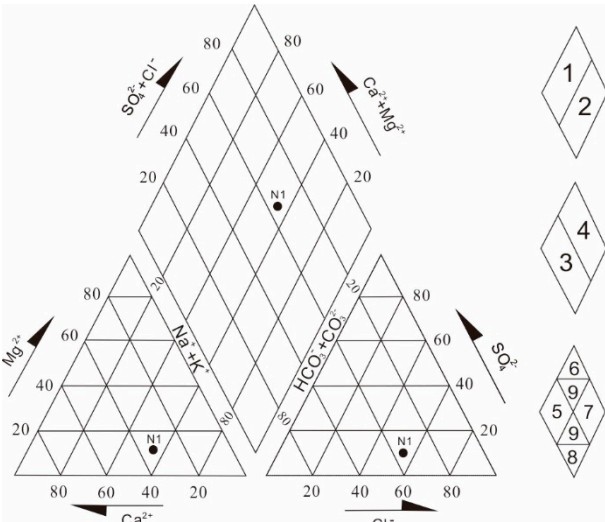

**Figure 7.** Piper diagrams of major ions in groundwater sample in the mud forest area.

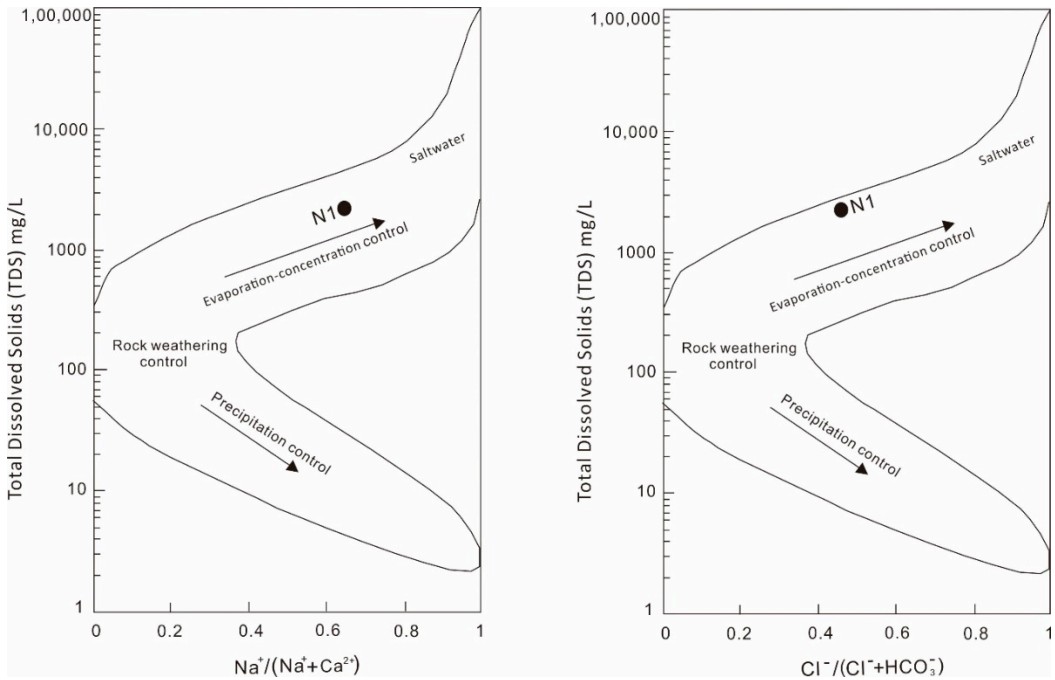

**Figure 8.** Plot of the major ions within the Gibbs model for groundwater in study area.

### 3.3. Soil Features

The development of subsurface erosion landform is closely related to soil properties. The soil that forms mud forest landform is a kind of loess-like soil (Figure 9). The results show that (a) all particles are smaller than 0.5 mm, (b) fine particle fraction (clay and silt) is more than three quarters, (c) a silt fraction (0.005–0.075 mm) accounts for the largest proportion, more than half. The soil was then classified as lean clay (CL) based on the Unified Soil Classification System (USCS) [57,58].

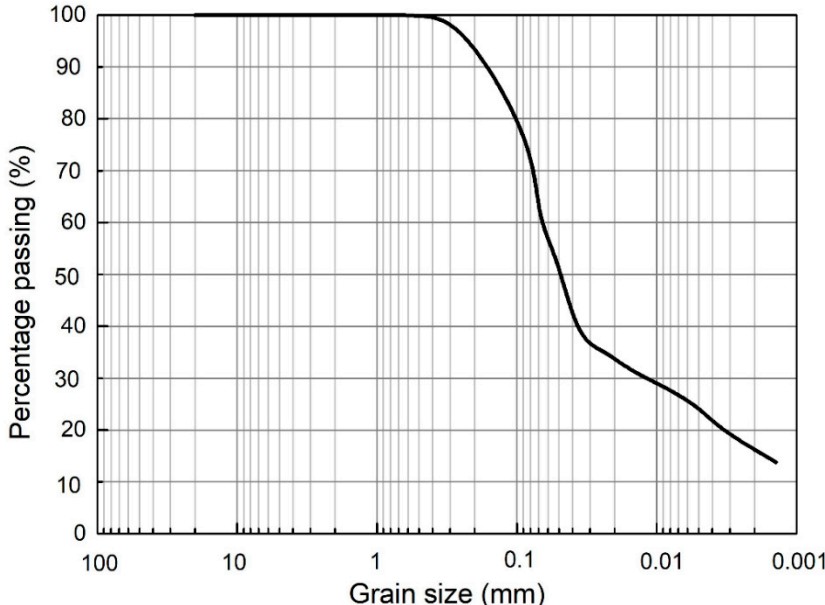

**Figure 9.** Grain size distribution of loess-like soil.

Table 5 shows the physical properties of loess-like soil sample, including particle size composition, plasticity index, liquid index, porosity, etc. Silt content accounts for 52%, which is the highest proportion. The plasticity index value of soil sample is low, and the liquid index is less than 0. The porosity of the soil sample is 57.46%.

**Table 5.** List of physical properties of loess-like soil sample.

| Item | Value |
|---|---|
| Clay content (%) | 24.42 |
| Silt content (%) | 52.75 |
| Sand content (%) | 22.83 |
| Natural density (g/cm$^3$) | 1.44 |
| Natural moisture content (%) | 13.23 |
| Liquid limit (%) | 26.78 |
| Plastic limit (%) | 18.26 |
| Liquidity index | −0.59 |
| Plasticity index | 8.52 |
| Soil classification | CL |
| Porosity (%) | 57.46 |

Chemical composition of loess-like soil can be seen in Table 6. From the oxide composition, the content of silicon dioxide is the highest, followed by alumina. This is similar to the composition of loess. Due to the high content of calcium oxide, magnesium oxide and potassium oxide, it proves that the soil formation environment is relatively dry and cold. The calculated silicic acid coefficient ($Ki = SiO_2/Al_2O_3$) of the sample is 7.93. The larger Ki value indicated that the loess-like soil was formed

when the climate was relatively arid. Therefore, the soil was formed in a dry and cold environment by the analysis of the oxide content of the sample.

**Table 6.** Chemical composition of loess-like soil (major element, in wt.%).

| SiO$_2$ | Al$_2$O$_3$ | Fe$_2$O$_3$ | FeO | CaO | MgO | K$_2$O | Na$_2$O | TiO$_2$ | P$_2$O$_5$ | MnO | LOI |
|---|---|---|---|---|---|---|---|---|---|---|---|
| 77.38 | 9.76 | 1.26 | 0.54 | 4.29 | 1.16 | 2.96 | 2.24 | 0.58 | 0.07 | 0.07 | 4.74 |

Clay dispersibility is a good indicator of the dispersion vulnerability of soil and, therefore, of the associated risks of soil erosion [57]. The soluble salt composition of soil sample investigated in the present study is given in Table 7, together with the calculated SAR, PS and TDS. The salt composition affects the properties of loess-like soil, especially dispersibility. Plotting of values presented in Table 7 in Sherard diagram clearly shows the zone of soil samples. All samples plot zone A (Figure 10). This indicates that the soil in mud forest area is dispersive.

**Table 7.** Soluble salt content (meq/L) and related parameters controlling clay dispersibility.

|  | Na$^+$ | K$^+$ | Ca$^{2+}$ | Mg$^{2+}$ | SO$_4{}^{2-}$ | Cl$^-$ | HCO$^-$ | TDS | PS | SAR |
|---|---|---|---|---|---|---|---|---|---|---|
| C1 | 7.844 | 0.014 | 1.896 | 0.301 | 1.623 | 4.011 | 4.865 | 10.055 | 78.011 | 7.484 |
| C2 | 7.452 | 0.011 | 1.513 | 0.228 | 1.733 | 5.255 | 3.586 | 9.204 | 80.965 | 7.987 |
| C3 | 8.041 | 0.010 | 1.577 | 0.285 | 1.137 | 5.550 | 4.254 | 9.913 | 81.116 | 8.334 |

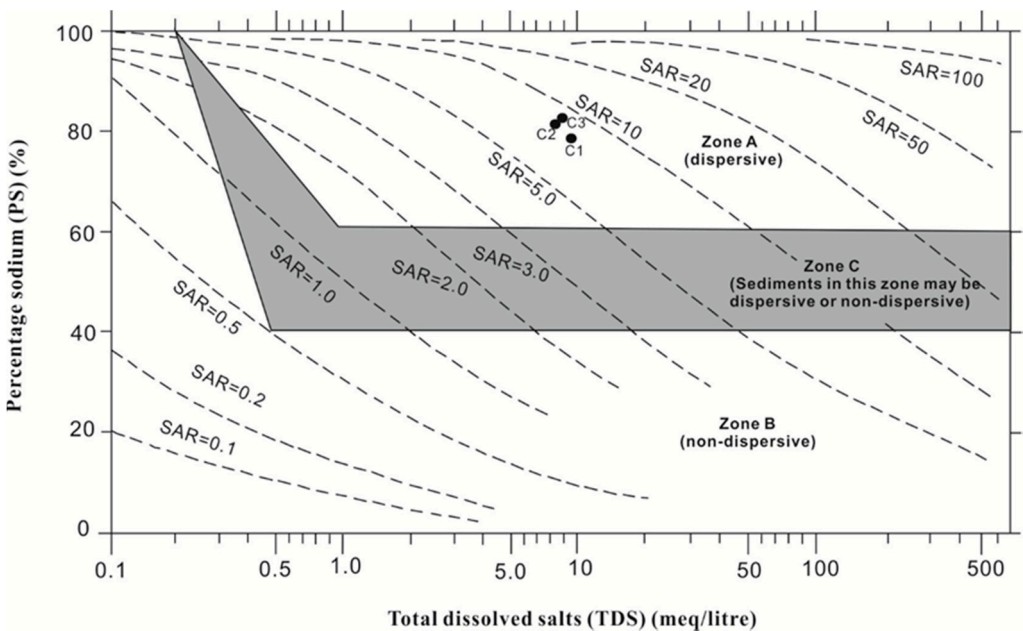

**Figure 10.** Relationships between clay dispersibility (susceptibility to colloidal dispersion) and salt composition (expressed through the PS, TDS and SAR parameters defined in Table 7).

### 3.4. Influence of Environmental Factors on Subsurface Erosion

Songnen Plain, a basin-like concave plain, provides a site for the formation of the mud forest landforms. Since the terrain is slightly concave, groundwater and precipitation tend to accumulate there. This, to some extent, aggravates the occurrence of subsurface erosion. During the Neoid period, gullies formed along the cutting of the earth's crust provided the possibility of erosion and subsurface erosion. Moreover, the uplift caused the inclined rise to the northwest, accompanied by new fault activities, which led to the asymmetry of landform between the east and west sides of the Dabusu Lake (Figure 11). Therefore, erosion and subsurface erosion on the east bank are more intense.

The asymmetry of the lake banks is not only the reason why the mud forest landscape began to occur, but also an important geological factor in the subsurface erosion that the mud forest is still experiencing.

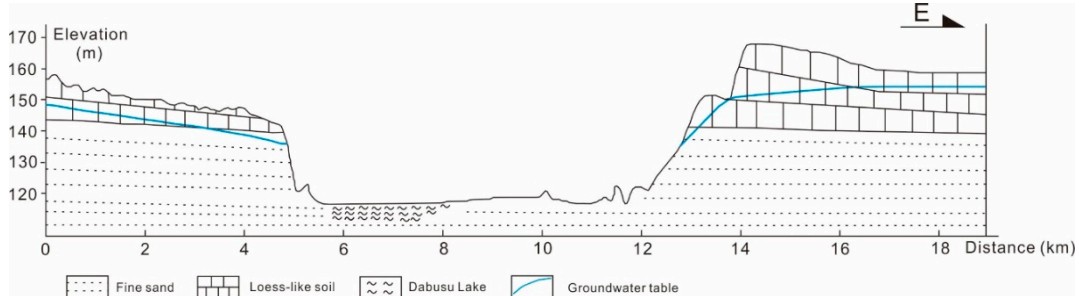

**Figure 11.** Profile of Dabusu Lake and its banks.

Due to the control and influence of neotectonic movement, the uplift of Xuezijing was active. In the late Pleistocene (about ten thousand years ago), there was a local uplift in Xuezijing, and the front edge of the second terrace formed a steep ridge more than 20 m high. The leading edge of the terrace floor was raised, showing an abnormal phenomenon that the front edge of the terrace slopes towards the back edge. The height difference between front edge and back edge was nearly 1 m. The terrace floor shows a north-south extension, with high in the middle, low on both sides. As the climate was dry and the ground rose again, the lake level dropped. Long-term, intense erosion and subsurface erosion occurred in lakeshore gullies.

Zhu and Liang [47] took sporopollen samples from the late Pleistocene lacustrine deposits and analyzed them. The results showed that the main sporopollen of woody plants were pinaceae and betulaceae, and the main sporopollen of herbaceous plants were artemisia and chenopodiaceae, ephedra and other drought-tolerant plant sporopollen also could be seen. This indicated that the Late Pleistocene climate was a dry and cold climate. This climate period corresponds to the Dali ice age in China. Fossils of fauna that lived in cold climates, such as mammoths and woolly rhinoceros, have been found, confirming a dry and cold climate. In this dry and cold environment, the lake water evaporates and condenses, increasing the salinity of the lake. The lake water with high salt content is deposited in direct contact with the sediments. This exacerbates the dissolution of soluble salts in loess-like soil. Thus, it is more prone to occur subsurface erosion.

Affected by the temperate monsoon climate, the study area is hot and rainy in summer and cold and dry in winter. Southwest winds prevail in the study area, with strong spring winds. The average annual wind speed is 4.1 m/s and the average annual maximum wind speed is 28.2 m/s [59]. Large amounts of clay and salt are blown toward the lakeshore. During the summer, some of the material is carried back into the lake by the wind. This ongoing wind cycle has a profound impact on molding the shape and physical structure of the lake basin. At present, the mud forest geological and geomorphic landscape is exposed to the wind, so the cutting and shaping of the lake basin and mud forest by wind force factors cannot be ignored. In addition, precipitation in summer is also a feature of this climate type. Annual precipitation is 400–500 mm [59]. In June, July and August, the precipitation reaches two-thirds of the annual precipitation. The average annual precipitation in summer is 359.9 mm [60]. Concentrated rainfall in a short period causes the exposed loess-like soil to be eroded. With the occurrence of rainfall infiltration and subsurface erosion, the geomorphology of mud forest changes and promotes the evolution and degradation of the mud forest landscape.

The formation and development of the mud forest geomorphic landscape is also closely related to the occurrence and movement of groundwater around it. Dabusu Lake and its surrounding areas are not only the center of precipitation and surface water collection in the Songnen alluvial plain, but also the center of groundwater collection. In addition, it is a confined water basin with multiple aquifers, which is rich in groundwater. Due to the uplift of the crust during the uplift period and the new faulting activities, the shallow groundwater was exposed. The Quaternary confined water may also

be exposed. So, the gully where the mud forest geological and geomorphic landscape is located is almost always filled with water. This provides a long-term underground water-rich environment for the formation of subsurface erosion geomorphology.

The buried groundwater level in the study area is shallow (Figure 11). The average groundwater level is 15.2 m. So the hydraulic gradient near the lakeshore is high. This leads to serious seepage damage of the soil. The subsoil of loess-like soil is a fine sand layer with good permeability. The water in the river is complementary through this highly permeable layer. Moreover, due to the high permeability of soil layer, groundwater has a higher velocity, and the decomposed soil particles can be carried away in time. As shown in Figure 12, infiltration of the precipitation converges with groundwater flow, continuously eroding loess-like soil, and over time, the mud forest landform landscape is formed. The content of sodium ion in water is higher, and it has the ability to disperse colloidal substance. At the same time, as this type of water circulates through the soil, it can increase the soluble salt content in the soil. While this is also an important reason for the degradation and destruction of the mud forest landscape.

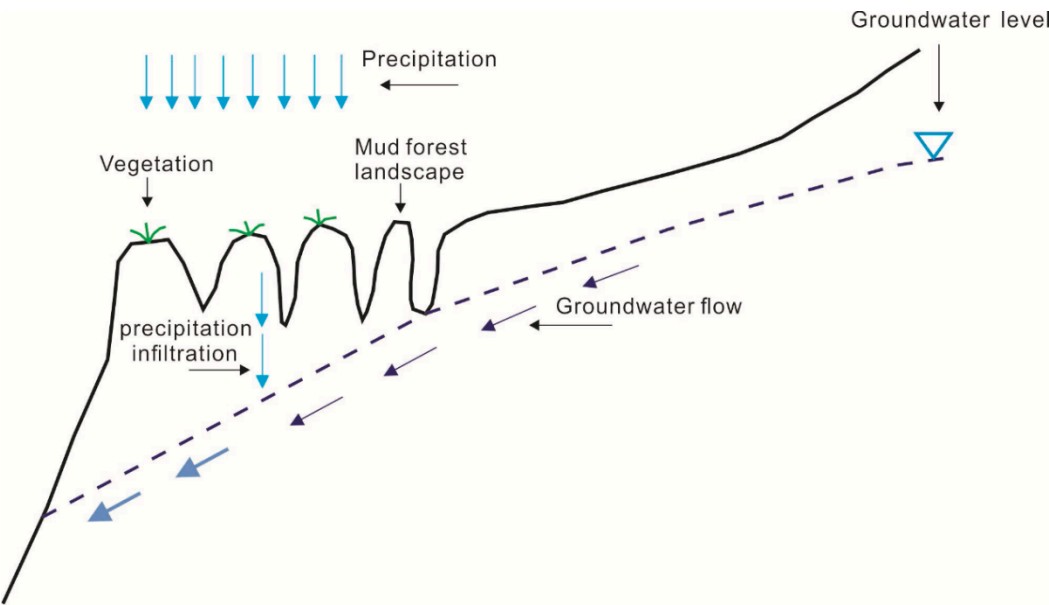

**Figure 12.** Groundwater flow and precipitation infiltration in mud forest area.

The results of the soil particle analysis tests showed that the soil sample contained the largest proportion of fine particles. While fine and very fine sand fractions were most important in controlling soil erosion [61], the high content of fine grained soil and weak hydrophilicity of the silt promoted the development of subsurface erosion in mud forest areas. Soil has no plasticity, more prone to subsurface erosion damage. Larger porosity of loess-like soil in mud forest area can promote the occurrence and development of subsurface erosion. Because the pores provide channels for the erosion products, accelerating the subsurface erosion process. The soil in mud forest area is dispersive (Figure 10). Thus, loess-like soil has a great propensity to produce a colloidal dispersion when groundwater flows through. After colloidal dispersed, small particles are carried along the pores by the water flow, forming cracks. Dispersive clays develop large pipes and erosion tunnels through rapid enlargement of small cracks and fissures as a consequence of the spontaneous dispersion of clays lining the fissure walls when these come in contact with water. This promotes subsurface erosion, thereby exacerbating the degradation of the mud forest landscape.

## 4. Conclusions

Mud forest landscape is a rare erosion landscape protected by the state because of its unique environmental features. In recent years, mud forest geological landscape began to degrade. In this paper, the evolution of the mud forest landscape was described, and the mechanisms for such evolution were analyzed.

After geological survey and analysis, mud forest landscape is the product of long-term geological processes. It is found that the mud forest landscape is in a dynamic evolution process. Therefore, it is named the dynamic evolution of the mud forest landscape into four stages in chronological order as infant stage, juvenile stage, youth stage and old stage, reflecting the evolution process of its formation to its extinction.

Therefore, it is discussed the factors that promote the subsurface erosion process in mud forest area and analyzed the influence of geological features, climate, groundwater and soil. It includes that: (1) The east and west banks are asymmetric, and the erosion of the east bank is more intense. Local uplift and lower lake levels lead to greater head differences; (2) The dry and cold paleoclimatic provides the environmental factor for the subsurface erosion, and the present arid-semiarid climate also facilitates the subsurface erosion; (3) There is abundant groundwater in the mud forest area, which provides a water-rich environment for the subsurface erosion. The groundwater is controlled by evaporation and concentration type, it provided abundant soluble salt for the soil in the cycle and promoted the occurrence of subsurface erosion; and (4) Loess-like soil has high content of silty grains, poor hydrophilicity, poor soil plasticity, and high porosity. It also has dispersibility, which makes it prone to subsurface erosion in contact with water. This is the root cause of subsurface erosion in mud forest areas.

**Author Contributions:** Data curation, X.R. and L.N.; Formal analysis, X.R., L.N. and Y.X. and C.D.; Funding acquisition, Y.X. and L.N.; Investigation, X.R., F.K. and Y.W.; Methodology, X.R., L.N. and T.Z.; Writing—original draft, X.R. and Y.H.; Writing—review & editing, Y.X and C.D. All authors have read and agreed to the published version of the manuscript.

**Funding:** This research received financial support from the Natural Science Foundation of China (grant no.41702300), (grant no.41502270), and (grant no. 41572254).

**Acknowledgments:** The authors are grateful for financial support from the National Natural Science Foundation of People's Republic of China (grant no.41702300), (grant no.41502270), and (grant no. 41572254). We wish to thank the testing science experiment center of Jilin University for supporting the experiment. We would also like to thank the Jilin Dabusu National Nature Reserve Administration for supporting our research work.

**Conflicts of Interest:** The authors declare no conflict of interest.

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
