# Peer review of "Characterization of Evolution Stages, Groundwater and Soil Features of the Mud Forest Landscape at Qian-an (China)"

_applsci, doi:10.3390/app10217427_

Round 1

Reviewer 1 Report

It is a very interesting work with important data about the factors that interfere on mud forest formation. The abstract and the introduction are good. However, in the Results and Discussion section there is some mixture with the Materials and Methods section. You should also enrich the Results and Discussion section with data on wind speed, precipitation, infiltration rate, ...

How many soil samples were taken and at what depth? How deep was the subsurface erosion assessed?

Have you observed and described visible signs of subsurface erosion in the field?

Line 247: Can you indicate the wind speed value?

Lines 253 and 254: Can you indicate the precipitation values here?

Line 254: What is the indicative value of heavy rainfall?

Line 255: Has the infiltration rate been assessed in the field? It would be very interesting to have data for this parameter.

Line 269: What is the average depth of the groundwater?

Line 273: Did you measure the speed of ground water? It would be very interesting to have data for this parameter.

Figures 9, 10 and 12 must appear first in the Materials and Methods section and must be explained in this section

Caption the black points in Figure 9 and 10

Standardize citations throughout the text

In Table 2, put the meaning of geologic symbols

In Tables 3 and 4 correct the word pH (lowercase p)

Author Response

Dear professor,

Thank you very much for your review of our article (Manuscript ID: applsci-963348). 

Pleased see the attachment.

Reviewer 2 Report

Line 73

Use the term „frost weathering“.

Line 77

Are you sure you want to use „landscape degradation“? It is about natural processes. Better to use „landscape transforming“ or „landscape evolution“.

Line 79

There is "subsurface flow". Geomorphologists use the term „interflow“.

Line 87-144, 2.1. Study area

You use USCS Soil Classification. USCS describe the texture and grain size of soli.
But to describe the soil please also use the Soil Taxonomy. With what soil we have to do? Horizons and Characteristics Diagnostic?

Line 93-94, Figure 1.

Complete the map scale.
Who is the author of the photos? Photo source?

Line 100-101

There is „The multi-year average precipitation is…“. It should be „Average annual precipitation is…“

There is „over two thirds of which is concentrated…“. It should be „Two-thrids oft the annual precipitation occurs…“

There is „The perennial average sunshine hours are 2900 hours.“ It should be „The average sunshine duration reaches 2.900 hours within a year.“

There is „Annual frost - free period is 145 days.“ It should be „The frost-free days are observed 145 days a year.“

Line 103

There is „This park has…“ It should be „In the park there…“

Line 105

Supplement the description of the slope with the difference in height.

Line 109

There is no justification as to why the Dabusu Lake is representative.

Linie 144-162

Relevant information is missing. What was the exact location where soil and water samples were taken? Amount, depth, sampling frequency?

Linie 171

There is „t0“. It should be „t01 and t02“

Linie 173

There is „t3“. It should be „t31 and t32“

Figure 5., Figure 6.

Who is the author of the photos?

Linie 200-202

Below terms that are not fit for scientific publication. They are acceptable in descriptions for tourists but in this article should be deleted or edited.

There is „…like the pyramids of ancient Egypt, and some look like the hump“. It should be „…like conical and humped forms“

There is „…is like a gallery of subsurface erosion landforms in the world.“ This description should be deleted.

There is „…is also the best viewing…“ It should be „…it is the most conspicuous…“

Figure 7.

What is the profile width? Describe the X axis.
Complete geographic directions.

Line 250

There is „…is facing the wind…“ It should be „…is exposed to the wind…“

Linie 252

Delete the word „concentrated”.

Linie 257

The name of this chapter should be „3.2.3. Hyrological Factors”

Linie 264

„…which is supplied by diving.” The sentence is incomprehensible.

Linie 268

There is „…precipitation infiltration…” It should be „…infiltration of the precipitation…“

Linie 316

It is unclear what the authors mean by „powder“? We do not use this word in the granulometric classification.

In conclusion, it is worth mentioning the need to protect this area. What is the forecast of the intensity of erosion processes in the context of climate change?

General comment

A thorough correction of the English text is required. The the text must be rewritten preferably by a native speaker. I only noted some examples of linguistic errors.

Author Response

Dear professor,

Thank you very much for your review of our article (Manuscript ID: applsci-963348).

Reviewer 3 Report

I think that the manuscript is quite well-written and needs rather technical than substantive comments. Please refer to the attached PDF and follow my comments. 

Author Response

(The authors gave the same response as above.)

Reviewer 4 Report

Authors describes landforms and groundwater and soil samples features with the aim to enhance the effect of subsurface erosion on the evolution of erosion landforms at Qian-an Mud Forest in China.

The topic is quite interesting and the four recognized evolution stages are overall well described. Whilst analysing both groundwater and soil samples is an important strength of the paper, such potential is not well exploited for the characterization of mud forest landscape in the study area.

Substantive or structural changes to the manuscript are required, particularly concerning data discussion. In particular, I warmly recommend to enrich and re-organize the subsections of "Results and Discussion" (detailed below) with the aim to allow the reader to follow all the phases of reasoning.

In the following, there are the main points on which I believe the authors need to supply more information or consider. I hope such considerations would be helpful in refining the manuscript, sometimes not easy to follow. For minor suggestions, please, see the attached pdf file. 

  1. Title

According to me, the title is not fully representative of the manuscript. I would suggest to modify the title in something similar to Characterization of evolution stages, groundwater and soil features of the mud forest landscape at Qian-an (China) 

  1. Abstract

Some repetitions occur and many refinements are annotated in the pdf file. 

  1. Material and methods: Data collection

Additional information concerning soil sample collection (L.147) should be provided. How many soil samples have been analysed? Have been the soils sampled at different heights within the same landform? How many samples have been collected for each landform evolution stage? 

  1. Results and discussion: section re-organization

The role of the geology and climate described at L.215-276 doesn’t derive from data analysed in this paper. For this reason, I suggest to move this part onward in the manuscript, as a general discussion after your own results have been presented.

In particular, an effort is required to Authors for re-arranging the subsections of “Results and discussion” by following a logical order similar to the following:

    1. Mud forest landscape features and evolution stage: for each  evolution stage, a description of landscape and landforms resulting from field survey is provided. Moreover, I suggest the Authors to combine figs. 4 and 6 in a single figure, by showing the scheme of each phase of evolution on the left side (fig. 4) and a corresponding significant picture (fig. 6) on the right side. An example is shown at L.169 of the attached file.
    2. Groundwater analyses: chemical characterization of collected samples
    3. Soil features: a description and interpretation of soil properties in the study area is required. It would be interesting to know if soil features vary across the four different evolution stages recognized by the Authors. In order to assess if the phase of landscape development influences soil features, I encourage the Authors to analyse samples collected in landforms with different evolution stages separately and discuss the findings accordingly.
    4. Overall discussion of the influence of environmental factors (climate, geology, groundwater and soils) on subsurface erosion as a consequence of both your own results and literature data. Further issues to consider are: is there any relationship between soil features and groundwater properties in the study area? Such a relationship varies with the evolution stage of the landscape?
  1. Clarity of the text

I warmly suggest the Authors to enhance the writing by an extensive English revision and avoiding repetitions and the use of personal pronouns (e.g. L.78, 79, 81, etc…).

Author Response

(The authors gave the same response as above.)

Round 2

Reviewer 1 Report

Standardize bibliographic citations in the text ( authors name or number)

Author Response

Dear professor,

Thank you very much for your review of our article (Manuscript ID: applsci-963348).

Thank you very much for your suggestions. We have revised the article accordingly, which significantly improves the quality of the work.

Reviewer 4 Report

Manuscript Number: APPLSCI-963348, SECOND REVIEW

Dear Authors,

Thank you for answering my comments and for accepting my suggestions. I particularly appreciate your effort to modify the structure of the “Results and discussion” section. After your corrections, the manuscript has been significantly improved however, a moderate English revision is still required.  You’ll find some annotations in the attached file.

Kind regards.

Author Response

(The authors gave the same response as above.)
